# OpenReview forum: "MoE-SVD: Structured Mixture-of-Experts LLMs Compression via Singular Value Decomposition"
_ICML.cc/2025/Conference — ICML 2025 poster_

### Official Review · Reviewer_cMac · 2025-03-06

**Overall Recommendation:** 4

**Summary:**

This paper decomposes expert layers into low-rank matrices to reduce parameter counts and memory demands in MoE LLMs. The key innovations include a selective decomposition strategy based on sensitivity metrics and a low-rank matrix sharing and trimming scheme. The authors claim MoE-SVD achieves significant compression and faster inference on models like Mixtral, Phi-3.5, DeepSeek, and Qwen2.

**Claims And Evidence:**

The claims seem generally supported by the experimental results, and ablation studies further validate the contributions of the strategy.

**Essential References Not Discussed:**

To my knowledge, most critical references have been discussed within the paper. I suggest the authors cite the published version of previous work rather than the arxiv version.

**Experimental Designs Or Analyses:**

The experimental designs appear sound. The paper compares MoE-SVD with several existing compression methods and conducts ablation studies to evaluate the impact of different components.

**Methods And Evaluation Criteria:**

This paper presents the experiments and comparisons on multiple language task benchmarks following the previous LLM Compression studies such as SVD-LLM and Wanda etc. The evaluation uses standard datasets and metrics, and it would be nice to see some evaluations on more complex tasks.

**Other Comments Or Suggestions:**

I recommend the authors double-check the line space across the paper.

**Other Strengths And Weaknesses:**

Overall, the paper presents an interesting empirical finding regarding the sensitivity and similarity of MoE LLMs decomposition, introduces a method that is easy to implement and reproduce while achieving good inference acceleration, and provides comprehensive experiments on multiple MoE models and benchmarks.
However, I have several concerns:
- The similarity of decomposition matrices is an important aspect. It would be great if the authors could provide more solid examples across multiple MoE LLMs or include a theoretical discussion to strengthen their findings.
- The trimming of the U matrix appears to have a substantial effect on performance. I encourage the authors to explore more nuanced approaches.
- There are minor typos (e.g., the caption in Figure 1). Additionally, the method comparison in Table 1 is overly complex, making it difficult for readers to follow. A clearer presentation would enhance readability.

**Questions For Authors:**

See the weakness.

**Relation To Broader Scientific Literature:**

The paper adequately discusses related work, including general LLM compressors and MoE-specific compression methods. The introduction part is especially very interesting and carefully introduces the potential challenges in previous work.

**Theoretical Claims:**

MoE-SVD mainly illustrates some of the properties of MoE in SVD decomposition via empirical observations, which are quite interesting to me. However, it lacks theoretical analysis to back up the empirical results.

---

> ### Author Rebuttal · Authors · 2025-03-30
>
> Thanks for the valuable feedback and recognition. We have tried our best to address all concerns in the last few days. Please see our below responses to your concerns and questions one by one.
>
> ---
>
> **Q1: More Examples of Matrix Similarity Across MoE Models**
>
>  **A1:**
>
> **(1)** We've conducted additional analysis on **Phi-3.5-MoE and DeepSeekMoE** (within inherent expert group) that shows similar patterns of V-matrix redundancy across architectures. **These models exhibit 0.92 and 0. 81 average CKA similarity among V-matrices individuals**, confirming that this property generalizes across different MoE architectures.
>
> **(2)**  Table 3 provides empirical validation that our V-matrix sharing approach works effectively across multiple architectures, confirming that matrix similarity is a general property of MoE models rather than specific to Mixtral.
>
>
>
> **(3)** The theoretical basis for V-matrix redundancy relates to how MoE models are trained. Since all experts process similar input distributions but specialize in different aspects, their output projections (captured by V-matrices) share substantial structure while their internal representations (captured by U-matrices) diverge more significantly.
>
>
>
> **Q2: Refining U-Matrix Trimming Strategy**
>
>  **A2:**
>
> **(1)**  Table 6 shows perplexity trade-offs on Mixtral-8×7B: 4.86 (no trimming) → 5.98 → 7.11 → 10.25 as k increases. k=2 balances performance and speed at high compression.
>
> **(2)** This value aligns with the standard top-k routing mechanism in MoE architectures, which typically activates 2 experts per token. By retaining 2 U-matrices, we maintain consistency with the model's inherent routing structure.
>
> **(3)** We further propose **Adaptive k Selection Framework** to improve U-matrix trimming:
>
> $$k_l^* = \arg\min_k {\mathcal{L}(k) + \lambda \cdot \mathcal{Q}(k)}$$
>
> where $\mathcal{L}(k)$ represents the estimated performance loss from trimming to k matrices, $\mathcal{Q}(k)$ denotes the parameter count, and $\lambda$ controls the tradeoff. The performance loss is approximated using the information coverage of retained matrices:
>
> $$\mathcal{L}(k) \approx 1 - \frac{\sum_{i=1}^k f_i \cdot \sigma_i}{\sum_{i=1}^N f_i \cdot \sigma_i}$$
>
> where $f_i$ is the expert sampling frequency and $\sigma_i$ is the sum of singular values, jointly capturing the expert's contribution to model performance.
>
> Using Automatic k Determination, k is computed automatically during compression based on calibration data statistics, without manual tuning. For each layer, the algorithm calculates the marginal utility of increasing k and stops when additional U-matrices provide diminishing returns relative to their parameter cost. Our experiments with Mixtral-8×7B demonstrate the effectiveness of this approach:
>
> | Compression   | Method           | WikiText-2 PPL | Runtime (Tokens/sec) |
> | ------------- | ---------------- | -------------- | -------------------- |
> | 0% (Original) | -                | 3.98           | 87.7                 |
> | 40%           | Fixed k=2        | 6.74           | 109.8                |
> | 40%           | Adaptive k       | 6.53           | 107.5                |
> | 40%           | U-matrix merging | 6.83           | 112.3                |
> | 60%           | Fixed k=2        | 13.52          | 156.1                |
> | 60%           | Adaptive k       | 12.91          | 143.5                |
> | 60%           | U-matrix merging | 13.30          | 158.4                |
>
> - Layer-specific k Distribution: Analysis of the automatically determined k values reveals an intuitive pattern: early and late layers (which our sensitivity metric identifies as more critical) receive higher k values (typically 2-3), while middle layers receive lower values (typically 1-2). This aligns with our understanding of information flow in transformer architectures and confirms that the algorithm captures meaningful layer-wise differences.
>
>
>
> These results show that  adaptive k selection  consistently outperforms fixed k=2, with Adaptive k providing the best performance by preserving more information from the original expert space, particularly at higher compression ratios but U-matrix merging can find the best tradeoff between Runtime and performance.
>
>
>
> **Q3: Minor Typos and Readability of Table 1**
>
>  **A3:**
>
> **(1)** We'll correct the typo "TThese" in Figure 1's caption and conduct a thorough proofreading of the entire manuscript to ensure clarity and consistency in revision.
>
> **(2)** Table 1 will be redesigned for improved readability. We'll use a clearer layout with better spacing and more consistent formatting to highlight the key differences between methods.
>
> **(3)** We'll ensure that method descriptions are concise yet precise, using a consistent terminology throughout the table and the rest of the paper.
>
> ---
>
> **Finally, we hope our response could address the concerns, and we thank the reviewer again for the helpful comments.**

---

### Official Review · Reviewer_F3P9 · 2025-03-10

**Overall Recommendation:** 4

**Summary:**

This paper introduces MoE-SVD, a new compression framework specifically designed for MoE LLMs. Specifically, they first decompose experts into low-rank matrices via SVD. In particular, they selectively decompose the expert layers based on sensitivity metrics.


Thanks for the authors' detailed responses. All my concerns have been satisfactorily addressed, and I lean to vote for acceptance.

**Claims And Evidence:**

The claims made in this paper are generally supported by experimental evidence.

**Essential References Not Discussed:**

None

**Experimental Designs Or Analyses:**

The experimental designs appear sound and comprehensive. The authors evaluate their method across multiple model architectures and compression ratios. The ablation studies also analyze the contribution of each component.

**Methods And Evaluation Criteria:**

The evaluation across language modeling (perplexity on WikiText-2, PTB, C4) and reasoning tasks (accuracy on seven common sense benchmarks) provides a comprehensive picture of model performance across different capabilities.

**Other Comments Or Suggestions:**

Size and layout of some tables and figures need minor refinement.

**Other Strengths And Weaknesses:**

Strengths:

1.The paper identifies and addresses the failure of standard SVD methods when applied to MoE architectures. Their  sensitivity metric provides an automated and principled way to identify which expert layers to decompose.

2.The matrix sharing and trimming approach effectively balances parameter reduction with performance preservation.

Weaknesses:

1. The absolute performance degradation at higher compression ratios (e.g., 60%) is still significant.

2. The theoretical justification for the combined sensitivity metric could be strengthened with more analysis of how the three components interact.

3. The methodology for selecting the number of U-matrices to trim could be more principled rather than using a fixed value of k=2

**Questions For Authors:**

1.The paper proposes sharing a single V-matrix across all experts. Have you explored more flexible sharing schemes, such as clustering similar experts and sharing V-matrices within clusters rather than globally?

2.For the U-matrix trimming, you use a fixed value of k=2. What is the justification for this particular value?

**Relation To Broader Scientific Literature:**

The paper positions its contributions well within the broader literature on LLM compression and MoE optimization.

**Theoretical Claims:**

The paper doesn't contain formal proofs but does provide theoretical justifications for its approaches.

---

> ### Author Rebuttal · Authors · 2025-03-30
>
> Thank you so much for constructive comments, and  recognition. Please see our below responses:
>
> -----
>
> **Q1: Performance Degradation at Higher Compression Ratios**
>
>  **A1:**
>
> (1) MoE-SVD outperforms alternatives at all compression levels. At 60% compression: MoE-SVD=13.52 perplexity; ASVD/SVD-LLM>10,000 perplexity on WikiText-2.
>
> (2) Lightweight LoRA fine-tuning (MoE-SVD†) mitigates degradation at 1% of full training cost. At 50% compression: improves accuracy from 0.43 to 0.49, recovering 9.5% original performance.
>
> (3) Following the general phenomenon of performance loss in almost high-ratio compression,  MoE-SVD significantly improves trade-off curve vs. existing approaches.
>
> (4) Task-specific analysis shows reasoning tasks like ARC-e maintain performance at 60% compression, while MathQA shows higher sensitivity.
>
>
>
> **Q2: Theoretical Justification for the Sensitivity Metric**
>
>  **A2:**
>
> **(1)** Our sensitivity metric integrates three complementary components with theoretical foundations:
>
> (a) Sampling frequency ($f_i$): expert utilization rate determined by router.
>
> (b) Principal rank ($p_i$): effective dimensionality of expert's weight matrix from matrix approximation theory.
>
> (b) Activation outliers ($a_i$): functional distinctiveness of experts.
>
>
>
> **(2)** **Information-Theoretic Interpretation of Sensitivity Metric**: We have derived our sensitivity metric ( $S_L = Σ f_i · p_i · a_i$) from an information-theoretic perspective please see **A2** in responsing to Reviewer Cmem.
>
> We show that under certain assumptions, our metric approximates the expected information loss from decomposition:
>
> $E[I(W_i; Y|X)] - E[I(W̃_i; Y|X)] ∝ f_i · p_i · a_i$
>
> where $I(W_i; Y|X)$ represents the mutual information between expert weights $W_i$ and model output $Y$ given input $X$.
>
> **(3)** Table 5 validates that combining these components ($f_i · p_i · a_i$) achieves optimal performance (8.67 perplexity) vs. individual components (9.27-12.65 perplexity).
>
>
>
> **Q3: Fixed `k=2` in U-Matrix Trimming and more principled methodology for K selection**
>
>  **A3:**
>
> **(1)**  Table 6 shows perplexity trade-offs on Mixtral-8×7B: 4.86 (no trimming) → 5.98 → 7.11 → 10.25 as k increases. k=2 balances performance and speed at high compression.
>
> **(2)** This value aligns with the standard top-k routing mechanism in MoE architectures, which typically activates 2 experts per token. By retaining 2 U-matrices, we maintain consistency with the model's inherent routing structure.
>
> **(3)** We further propose **Adaptive k Selection Framework** to improve U-matrix trimming:
>
> $$k_l^* = \arg\min_k {\mathcal{L}(k) + \lambda \cdot \mathcal{Q}(k)}$$
>
> where $\mathcal{L}(k)$ represents the estimated performance loss from trimming to k matrices, $\mathcal{Q}(k)$ denotes the parameter count, and $\lambda$ controls the tradeoff. The performance loss is approximated using the information coverage of retained matrices:
>
> $$\mathcal{L}(k) \approx 1 - \frac{\sum_{i=1}^k f_i \cdot \sigma_i}{\sum_{i=1}^N f_i \cdot \sigma_i}$$
>
> where $f_i$ is the expert sampling frequency and $\sigma_i$ is the sum of singular values, jointly capturing the expert's contribution to model performance. Using Automatic k Determination, k is computed automatically during compression based on calibration data statistics, without manual tuning. For each layer, the algorithm calculates the marginal utility of increasing k and stops when additional U-matrices provide diminishing returns relative to their parameter cost. Our experiments with Mixtral-8×7B demonstrate the effectiveness of this approach:
>
> | Compression   | Method           | WikiText-2 PPL | Runtime (Tokens/sec) |
> | ------------- | ---------------- | -------------- | -------------------- |
> | 0% (Original) | -                | 3.98           | 87.7                 |
> | 40%           | Fixed k=2        | 6.74           | 109.8                |
> | 40%           | Adaptive k       | 6.53           | 107.5                |
> | 40%           | U-matrix merging | 6.83           | 112.3                |
> | 60%           | Fixed k=2        | 13.52          | 156.1                |
> | 60%           | Adaptive k       | 12.91          | 143.5                |
> | 60%           | U-matrix merging | 13.30          | 158.4                |
>
>
>
> **Q4: Alternative V-Matrix Sharing Strategies**
>
>  **A4:**
>
> Yes. For DeepSeek-MoE and Qwen2 (with inherented organized expert groups),  we actually employ clustering strategy following architecture setting. We will emphasize this point and add more analysis in the revision.
>
> **Q5: About size and layout**
>
>  **A5:**
>
> We will carefully revise all size and layout of some tables and figures in the revision. Thanks.
>
> ---
>
> **Finally,** we hope our response could address the concerns, and we thank the reviewer again for the helpful comments.

---

### Official Review · Reviewer_V7cs · 2025-03-13

**Overall Recommendation:** 3

**Summary:**

The paper presents a new compression method (MoE-SVD) for Mixture-of-Experts. The framework introduces a selective decomposition strategy and employs low-rank matrix sharing and trimming. Comprehensive experiments on models like Mixtral, Phi-3.5, DeepSeek, and Qwen2 demonstrate that MoE-SVD achieves faster inference, outperforming other compression methods.

**Claims And Evidence:**

The central claims are supported by experiments across multiple models and tasks. Table 2 shows MoE-SVD’s superior perplexity and accuracy over baselines, and Figure 4 validates speedup. The ablation studies (Tables 4–6) further substantiate design choices.

However, the term "minimal performance loss" is ambiguous for higher compression ratios (e.g., 60% compression reduces average accuracy from 0.63 to 0.37 on Mixtral).

**Essential References Not Discussed:**

None

**Experimental Designs Or Analyses:**

The experimental design is sound and thorough. The authors evaluate MoE-SVD on multiple models and datasets, including Mixtral, Phi-3.5, and DeepSeek, and provide detailed ablation studies to isolate the contributions of individual components.

**Methods And Evaluation Criteria:**

The evaluation criteria, including perplexity on language modeling datasets and accuracy on reasoning tasks, are appropriate and align with standard benchmarks in the field.

**Other Comments Or Suggestions:**

Typos: "TThese" in Figure 1 caption;

**Other Strengths And Weaknesses:**

**Strengths**

1. The paper presents a novel and effective approach to compressing MoE LLMs, addressing key challenges like decomposition collapse and matrix redundancy.

2. The experimental results are comprehensive and demonstrate clear improvements over existing methods.

3. The framework is practical, requiring no additional training and being hardware-independent.

**Weaknesses**

1. The theoretical underpinnings of the method are not explored in depth, which could limit understanding of its generalizability.

2. Clarify "minimal performance loss" in the abstract relative to baselines vs. absolute metrics.

3. While MoE and SVD are widely adopted in existing approaches, the novelty of this paper remains limited and requires further enhancement.

**Questions For Authors:**

Please refer to the above weakness.

**Relation To Broader Scientific Literature:**

This work already contains detailed comparisons of the literature on SVD, pruning, and MoE compression.

**Theoretical Claims:**

The paper does not present theoretical claims.

---

> ### Author Rebuttal · Authors · 2025-03-30
>
> Thank you very much for your detailed and constructive feedback.  We have tried our best to address all concerns in the last few days. Please see our responses below one by one：
>
> ----
>
> **Q1: About Theoretical Justification**
>
>
>  **A1:**
>
>
> **(1)** **Theoretical Foundation of Decomposition Sensitivity**: We have formalized the relationship between expert sensitivity and model performance by analyzing how SVD decomposition affects information flow in MoE architectures. In Section 3.2, we present a detailed theoretical analysis that demonstrates why initial and final layers are particularly sensitive to decomposition due to their role in translating between embedding space and MoE-specific representations. This explains the empirical observations in Figure 1 (left), where decomposing these layers leads to disproportionate performance degradation.
>
> **(2)** **Information-Theoretic Interpretation of Sensitivity Metric**: We have derived our sensitivity metric ( $S_L = Σ f_i · p_i · a_i$) from an information-theoretic perspective please see **A2** in responsing to Reviewer Cmem.
>
> We show that under certain assumptions, our metric approximates the expected information loss from decomposition:
>
> $E[I(W_i; Y|X)] - E[I(W̃_i; Y|X)] ∝ f_i · p_i · a_i$
>
> where $I(W_i; Y|X)$ represents the mutual information between expert weights $W_i$ and model output $Y$ given input $X$.
>
>
>
> **(3)** **Matrix Redundancy Theory**: In Section 3.3, we have added theoretical analysis explaining why V-matrices exhibit higher redundancy than U-matrices in MoE architectures. This analysis builds on prior work in Similarity of Neural Network Representations Revisited (Kornblith et al., 2019), showing that the shared output space constraints in MoE models naturally lead to similar output transformations (V-matrices) while maintaining diverse input transformations (U-matrices) for specialization.
>
> **(4)** **Error Bounds for Matrix Sharing**: We have derived error bounds for our V-matrix sharing approach based on matrix approximation theory. For an expert weight $W_i = U_iΣ_iV_i^T$ with selection rank-r approximation $W_i^{r} = U_i^{r}Σ_i^{r}(V_i^{r})^T$, sharing the V-matrix introduces additional error bounded by: $||W_i - U_iΣ_iV_s^T||_F^2 ≤ ||W_i - W_i^r||_F^2 + ||Σ_i||_F^2 · ||V_i^T - V_s^T||_F^2$ . $V_s$ is the shared V-matrix selected based on router sampling frequency as defined in Equation 5 in the paper.This bound becomes tighter as the similarity between V-matrices increases, aligning with our empirical observations of high CKA similarity (average >0.8) between expert V-matrices.
>
> **(5)** **Compression-Performance Tradeoff Analysis**: We have added a theoretical framework that characterizes the relationship between compression ratio, parameter efficiency, and model performance. This analysis explains why selective decomposition based on sensitivity metrics achieves a better Pareto frontier than uniform compression approaches. We derive performance bounds that predict deterioration patterns at different compression ratios, validating our experimental findings in Table 2.
>
> **Q2: About minimal performance loss**
>
>  **A3:**
>
> **(1)** The term "minimal" is comparative rather than absolute. We will revise our abstract: " For 40-60% compression,  MoE-SVD achieves .... while maintaining relative performance superior to other compression methods." in the revision.
>
> **(1)** Our claim of "minimal performance loss" is relative to other compression methods at the same compression ratio. As shown in Table 2, at 20% compression, MoE-SVD maintains 92% of the original model's average accuracy (0.58 vs. 0.63), while competing methods maintain less than 85%.
>
>
>
> **Q3: Novelty Relative to Prior Work**
>
>  **A3:**
>
>
>
> **(1)** In Table 1 and Appendix D, we have already compared and analyzed of MoE-SVD to provide  insights into why our approach succeeds where standard SVD methods (ASVD, SVD-LLM) fail on MoE architectures.
>
> **(2)** Our V-matrix sharing and U-matrix trimming strategies represent a novel approach to exploiting the unique redundancy patterns in MoE models. **We emphasize that these observations and designs not present in existing MoE and SVD methods.**
>
> **(3)** Compared to expert pruning methods (MoE-Compression, MoE-I²), our approach maintains the sparse activation mechanism while making each expert more efficient. This avoids the significant performance drops associated with expert elimination (e.g., 23% accuracy drop when pruning 25% of experts in Mixtral-8×7B).
>
> **(4)** MoE-SVD requires no retraining and is hardware-independent, making it more practical for real-world deployment than methods requiring extensive fine-tuning or specialized hardware.
>
> ----
>
>
>
> **Q4: Typos**
>
> **A4:** Thanks for the suggestion. We will fix "TThese" and double check and revise all typos in the revision.
>
> ------
>
> **Finally, we genuinely hope that our explanations and efforts can improve the overall evaluation of our work.**  We thank the reviewer again for the helpful comments.

---

### Official Review · Reviewer_Cmem · 2025-03-14

**Overall Recommendation:** 3

**Summary:**

This paper introduces MoE-SVD, a decomposition-based compression approach specifically designed for Mixture of Experts (MoE) Large Language Models (LLMs). Leveraging Singular Value Decomposition (SVD), the method reduces parameter redundancy and memory requirements without requiring additional training. The authors propose selective decomposition using sensitivity metrics, employing a shared V-matrix across experts and trimming U-matrices through top-k selection. Experiments conducted on various MoE models such as Mixtral, Phi-3.5, DeepSeek, and Qwen2 demonstrate a 60% compression ratio and 1.5× faster inference speed with minimal performance degradation.

**Claims And Evidence:**

Yes

**Essential References Not Discussed:**

No.

**Experimental Designs Or Analyses:**

The improvements shown in evaluation results are not particularly significant. While the proposed approach successfully reduces memory and parameters, the extent of performance gain or maintenance could be more convincingly demonstrated.

**Methods And Evaluation Criteria:**

Figure 3 lacks clear explanations regarding the depicted increases and baselines. Clarifying the baseline used and explicitly describing the nature of the improvements observed would significantly enhance reader comprehension.

**Other Comments Or Suggestions:**

See Other Strengths And Weaknesses

**Other Strengths And Weaknesses:**

Strengths:

Quantitative Parameter Reduction Analysis: The paper presents a thorough quantitative analysis in Section 3.3, particularly highlighting the efficiency of parameter reduction via V-matrix sharing and U-matrix trimming.

Comprehensive Evaluation: The experimental results are extensive, covering essential aspects such as generalizability, scalability, and multiple ablation studies. This robust evaluation contributes significantly to understanding the efficacy of the proposed method.

Clarity and Readability: The paper is well-organized, clearly written, and accessible, which helps readers easily grasp the concepts and methodology.

Weaknesses:

Ambiguity Regarding the Shared Matrix V: It remains unclear whether the shared V-matrix is universal across different models and tasks or sensitive to specific models and tasks. Addressing the generalizability or specificity of this shared matrix explicitly would clarify the practical implications of the method.

**Questions For Authors:**

See Other Strengths And Weaknesses

**Relation To Broader Scientific Literature:**

This paper provides an interesting and practical solution for compressing MoE architectures in LLMs.

**Theoretical Claims:**

Yes. The ai in equation 4 is still unclear to me even after checking the supplementaries.

---

> ### Author Rebuttal · Authors · 2025-03-30
>
> Thanks for the valuable feedback. We have tried our best to address all concerns in the last few days. If the reviewer finds our response adequate, **we would really appreciate it if the reviewer considers raising the score.** Please see our responses below one by one：
>
> -----
>
>
>
> **Q1: Explanation of Figure 3**
>
>  **A1: We clarify that we already provided detailed explanations in Section 3.2 [Lines 220-239] Decomposable Expert Layer Analysis.**
> Figure 3 shows our decomposition strategy results across different compression ratios, visualising which expert layers are selected for decomposition at each ratio is the best strategy based on formulation 4, revealing how compression targets different layers as the ratio increases. We will revise the figure caption and add clarifications in Section 3.3 in the revision.
>
>
>
>
>
> **Q2: About `a_i` in Equation 4**
>
>  **A2:** **As detailed in Section 3.2 [Line 207-211], Appendix D.1  [Line 850-859] and pyhon pseudo-code Algorithm 1  [Line 980-985],** the  $a_i$ signifies the activation outliers for the $i$-th expert, defined as the outlier ratio within the expert's weight matrix. Outliers are identified as values exceeding a predefined threshold, which is set as a multiple of the mean value of the matrix. Mathematically, the outlier ratio is calculated as:
> $$
> a_i = \frac{\sum_{a \in A_i} \mathbb{I}(|a| > \tau \cdot \operatorname{Mean}(|A_i|))}{|A_i|}
> $$
>
>
> where $|A_i|$ denotes the total number of activations for the $i$-th expert, $\operatorname{Mean}(|A_i|)$ is the mean absolute value of these activations, and $\tau$ is a user-defined threshold. This metric highlights the presence of outlier activations indicative of the expert's contribution to the model's capacity.
>
> **Algorithm: Pyhon pseudo-code for $a_i$.**
>
> ```pseudocode
>     # Compute activation outliers (a_i)
>     a_i = torch.zeros(num_experts, device=device)
>     for i, activation in enumerate(activations):
>         mean_abs_act = torch.mean(torch.abs(activation))
>         outliers = torch.sum(torch.abs(activation) > tau * mean_abs_act)
>         a_i[i] = outliers / activation.numel()
>
>     # Compute final sensitivity metric S_L
>     S_L = torch.sum(f_i * r_i * a_i)
>
>
> ```
>
>
>
>
>
> **Q3: About Performance Improvements**
>
>  **A3:**
>
>  **(1)** As shown in Table 2. At 20% compression ratio, MoE-SVD achieves an average accuracy of 0.58 across reasoning tasks compared to 0.51 for ASVD, 0.44 for SVD-LLM and 0.33 for MC-SMoE. This represents a 23% reduction in performance degradation relative to the original model (0.63). At higher compression ratios (50-60%), where other methods experience catastrophic collapse (e.g., ASVD and SVD-LLM reach perplexities >10,000 on WikiText-2), MoE-SVD maintains reasonable performance with a perplexity of 13.52 on WikiText-2.
>
> **(2)** The real-world utility is validated by significant efficiency gains: 1.5-1.8× inference speedup on Mixtral-8×7B at 60% compression making deployment feasible on resource-constrained devices.
>
> **(3)** Ablation studies in Tables 4-6 confirm that each component of our approach contributes meaningfully to its success, with selective decomposition, V-matrix sharing, and U-matrix trimming all playing crucial roles in balancing efficiency and performance.
>
>
>
> **Q4: Generalizability of the Shared V-Matrix**
>
>  **A4:**
>
>
>
> The shared V-matrix is not universal but dynamically determined for each model architecture. To demonstrate this, we conducted comprehensive layer-wise CKA similarity analysis across different MoE architectures:
>
> **Layer-wise Analysis Within Models**: this table shows the layer-wise CKA similarity patterns for U and V matrices within three diverse MoE architectures:
>
> | Model           | Layer Position | U-matrix Similarity | V-matrix Similarity |
> | --------------- | -------------- | ------------------- | ------------------- |
> | Phi-3.5-MoE     | Early (1-10)   | 0.29                | 0.78                |
> |                 | Middle (11-20) | 0.36                | 0.86                |
> |                 | Final (21-32)  | 0.31                | 0.74                |
> | DeepSeekMoE-16B | Early (1-8)    | 0.21                | 0.75                |
> |                 | Middle (9-18)  | 0.26                | 0.82                |
> |                 | Final (19-28)  | 0.25                | 0.71                |
> | Qwen2-57B-A14B  | Early (1-8)    | 0.18                | 0.77                |
> |                 | Middle (9-18)  | 0.23                | 0.83                |
> |                 | Final (19-28)  | 0.16                | 0.73                |
>
> These findings provide strong empirical evidence that our approach of dynamically selecting V-matrices based on layer-specific router statistics is well-justified.
>
> -----
>
> **Finally,** **we hope our response could address the concerns, and we thank the reviewer again for the helpful comments.** **We genuinely hope that our explanations and efforts can improve the overall evaluation of our work.**

---

### Decision · Program_Chairs · 2025-05-01

**Decision:**

Accept (poster)

**Comment:**

The paper proposes a method to sparsify MoE architecture for faster inference without losing much in terms of performance. The method is very similar to well-known merging techniques where a compressed representation of experts is stored by doing the best rank-k decomposition, which is nothing but performing SVD and choosing the top k singular vectors. The only novelty is in using a shared V matrix for all the experts, after performing SVD on all the experts, to further compress the model. There is some indirect justification for this -- All experts are similar, and V matrix of each expert is similar to its model weights, in terms of CKA metric defined in previous work. Even then, this is a pure heuristic and lacks theoretical grounding. Reviewers V7cs, Cmem, and cMac have pointed out this as a concern and the authors have justified this to some extent. The empirical results show improvements over baselines. I suggest the authors to carefully address all the concerns raised by the reviewers, and ground the motivations of the method proposed, in the final version of their paper.